# Integrating a UAV System Based on Pixhawk with a Laser Methane Mini Detector to Study Methane Emissions

**Timofey Filkin [1], Iliya Lipin [2] and Natalia Sliusar [1,\*]**

[1] Environmental Protection Department, Perm National Research Polytechnic University, 614000 Perm, Russia; filkin-t@yandex.ru

[2] FPV Perm, 614000 Perm, Russia; ilipin@yandex.ru

\* Correspondence: nnslyusar@gmail.com

**Abstract:** This article describes the process of integrating one of the most commonly used laser methane detectors, the Laser Methane mini (LMm), and a multi-rotor unmanned aerial vehicle (UAV) based on the Pixhawk flight controller to create an unmanned aerial system designed to detect methane leakages from the air. The integration is performed via the LaserHub+, a newly developed device which receives data from the laser methane detector, decodes it and transmits it to the flight controller with the protocol used by the ArduPilot platform for laser rangefinders. The user receives a single data array from the UAV flight controller that contains both the values of the methane concentrations measured by the detector, and the co-ordinates of the corresponding measurement points in three-dimensional space. The transmission of data from the UAV is carried out in real time. It is shown in this project that the proposed technical solution (the LaserHub+) has clear advantages over not only similar serial commercial solutions (e.g., the SkyHub complex by SPH Engineering) but also experimental developments described in the scientific literature. The main reason is that LaserHub+ does not require a deep customization of the methane detector or the placement of additional complex devices on board the UAV. Tests using it were carried out in aerial gas surveys of a number of municipal solid waste disposal sites in Russia. The device has a low cost and is easy for the end user to assemble, connect to the UAV and set up. The authors believe that LaserHub+ can be recommended as a mass solution for aerial surveys of methane sources. Information is provided on the approval of LaserHub+ for aerial gas surveys of a number of Russian municipal waste disposal facilities.

**Keywords:** unmanned aerial vehicles; methane emissions; methane leakage detection; tunable diode laser absorption spectroscopy

## 1. Introduction

Nowadays, the task of identifying and describing sources of methane emissions poses a significant challenge for many sectors of industry, agriculture and urban economics. Its relevance is connected not only to the current climate agenda and the need to address global warming, but also to the fact that methane is a major pollutant, especially when it comes to the formation of toxic ground-level ozone, the emissions of which must be kept in check [1]. Control and assessment of methane emissions is necessary not only for calculating the environmental damage already caused, but also for planning measures to reduce current and future emissions. Despite increased interest in the topic of methane leakage detection, researchers have noted that there continues to be a significant technological gap with regard to monitoring systems capable of effectively detecting leakages when they occur and determining their location with the accuracy necessary for repair crews [2].

Since many objects, actual or potential sources of methane emissions, are inconvenient for on-site studies due to factors such as size, extension, inaccessibility or other

dangerous and adverse effects, there is understandable interest among stakeholders in developing remote methods to detect and measure methane emissions. The results of review studies have repeatedly indicated that small (<20–25 kg) unmanned aerial vehicles (UAVs) are a very promising method for remotely conducting operational surveys of objects ranging in size from several tenths of a hectare to a square kilometer [3–5]. UAVs make it possible to carry out surveys with high spatial and temporal resolution, and to take measurements close to the source [2]. Over the past 15 to 20 years, significant efforts have therefore been made to improve unmanned systems for studying gas emissions and chemical pollution in the atmosphere. Projects thus far have focused on the following areas: modernizing the UAVs themselves (e.g., increased flight time, improved communication channels with the ground control station), making the payload smaller and more compact for placement on board the UAV (e.g., more advanced analyzers or samplers) and developing new ways to integrate onboard analyzers and UAVs into a single unmanned aerial system (UAS).

The last of these three areas of focus is especially important because a single UAS generates at least two data streams: flight information recorded by the UAV flight controller, and information about events involving the payload (e.g., taking measurements, air sampling). The ability to subsequently process the experimental field data collected, not to mention the quality thereof, depends greatly on how accurately these information flows were coordinated. In particular, it is crucial that measurements made by the onboard analyzer (or sampler in some cases) be georeferenced accurately, ideally on board the UAV in order to obtain a single data stream for processing.

Another problem that arises when developing UAS is how to ensure the replicability of the proposed solutions and their availability for the end user. Off-the-shelf commercial products are usually the most convenient, but their high price may encourage users to look for free or low-cost open access solutions. Thus, when analyzing what is available on the market for unmanned methane leakage detection systems, we can see that many solutions are based on commercial UAVs, mainly from the company SZ DJI Technology Co., Ltd. (Shenzhen, China), which produces the DJI Matrice series. Furthermore, specialized commercial software is often required to work with these technologies. On the other hand, custom-made drones built with a Pixhawk autopilot based on the ArduPilot software package are also widely used around the world [3,4,6]. ArduPilot is the most common and inexpensive platform but also a highly versatile one with an open architecture and open-source code for creating UAVs. Industry-specific studies have highlighted the importance of open source for both software and hardware as it allows researchers to design unmanned systems and plan surveys that precisely meet their unique needs [6]. Furthermore, there are promising opportunities for the widespread adoption of aerial gas survey systems that are cheaper than those available on the market today.

The purpose of this research is to illustrate the potential for simplifying and reducing the cost of using UAS to study methane emissions. As an example, the authors chose to focus on a system equipped with laser detectors, which make up a wide class of onboard methane analyzers. The promise of this technology, according to the authors, is associated with its capacity to facilitate the transfer of data from the onboard analyzer to the UAV autopilot, and then onwards from the autopilot to the ground control station. This can, in turn, be implemented on the technological platform Pixhawk using ArduPilot software, which is cheaper than its commercial counterparts. The main task of this project was to develop software and hardware for data transfer from the onboard detector to the UAV autopilot, and to establish procedures for synchronizing these data on methane concentration measurements with other data from the flight controller, in particular, the data from the built-in receiver, using a global navigation satellite system (GNSS-receiver). The results of this project indicate that is quite feasible to eliminate the need for additional programs and the added cost of intermediary devices. These results will be confirmed with further field research.

## 2. Materials and Methods

### 2.1. Selection of a Sensor for Methane Detection

There are three approaches that are most commonly implemented in UAV-based studies of methane [5,7]:

1.  Atmospheric sampling using onboard equipment followed by sample analysis with surface instruments;
2.  Analysis of air samples in real time by pumping them into a long tube connected to a ground analyzer;
3.  Real-time measurements taken directly with onboard instruments.

The last approach was determined by the researchers to be the most preferable [5].

All onboard analyzers designed for quantitative studies of methane emissions can be divided into two large groups: compact chemical sensors (usually electrochemical sensors) and high-precision optical analyzers [8]. Other types of devices that use optical gas imaging technology (e.g., OGI cameras) were not considered in this project because they are intended for qualitative research tasks, such as leakage visualization. That said, another predominant group of optical analyzers can be broadly categorized as "methane laser detectors", which use laser radiation and are based on the principles of absorption spectroscopy [5]. According to a number of recent review papers devoted to the study of using UAVs for methane emissions measurement, it can be concluded that in practice, laser detectors are more commonly integrated into UAVs compared not only to other optical analyzers (e.g., non-dispersive infrared sensors), but also to electrochemical sensors [5,9,10].

The widespread practice of integrating laser methane detectors into UAS can be explained by the fact that these detectors lack a number of shortcomings with regard to sensitivity that are present in electrochemical sensors (e.g., limited selectivity, vulnerability to meteorological changes). Moreover, their readings are not affected by the down-wash airflow created by the propellers of multi-rotor UAVs, which are more common than aircraft-type UAVs in methane studies. An additional advantage of laser detectors is their ability to evaluate the integral concentration of methane in the air column under the UAV, in other words, to obtain information about the entire surface air column in one measurement taken above a given point on the earth's surface.

Airborne laser methane detectors tend to use frequency-tunable pulsed diode lasers and are accordingly often referred to as TDLAS sensors (tunable diode laser absorption spectroscopy). These detectors emit laser radiation at a frequency tuned to the absorption characteristics of methane molecules. The radiation, passing through the layer (column) of air containing methane, reflects from an obstacle and partially returns to the device. Then, the degree of radiation absorption is calculated (i.e., how much energy the medium has absorbed) and the methane density in the gas layer along the optical path of the laser from the device to the object is determined (usually measured in ppm*m). The advent of these cost-efficient and lightweight lasers has led to a significant leap forward in the development of UAS for emissions monitoring [4]. Only in the last two or three years, with developments in the field of sensor miniaturization, has it become possible to use other laser detectors that would previously have been too bulky for UAS, such as those based on the principles of cavity ring-down spectroscopy (CRDS) [10].

In the course of this project focused on using UAVs with laser methane detectors, mainly in the fields of public utilities and natural gas transmission, it became clear that among the most commonly used devices are those manufactured by GASTAR Co., Ltd. (Japan, Yamato), especially the Laser Methane mini (LMm) detector. The LMm was developed in 1992 by Tokyo Gas Engineering Solutions (Japan, Tokyo) as a handheld portable detector for ground inspections [11]. In 2013, it became commercially available [4] and was then integrated into a variety of aerial remote-sensing systems, including the LMC-G2-DL made by JSC Pergam-Engineering (Russia, Moscow) and the mdTector1000 CH4 by Microdrones (Germany, Siegen). To date, the LMm is one of very few commercially

available methane sensors [4] that feature a fairly low total weight, a wide measurement range (from 1 to 50,000 ppm-m), a high sensitivity capacity and a high measurement speed (10 Hz). Consequently, the LMm was chosen as a representative sample of onboard analyzers to be integrated into UAS for methane leakage detection. In recent years, Tokyo Gas Engineering Solutions has developed the Laser Falcon detector, based on the LMm and designed specifically for UAVs, as well as a modified version of the LMm—the Laser Methane Smart. Unfortunately, it was not possible to find examples in the scientific literature of studies where these newer detector models were used. Figure 1 indicates where to place the selected analyzer for methane leakage detection in the full range of UAS payloads.

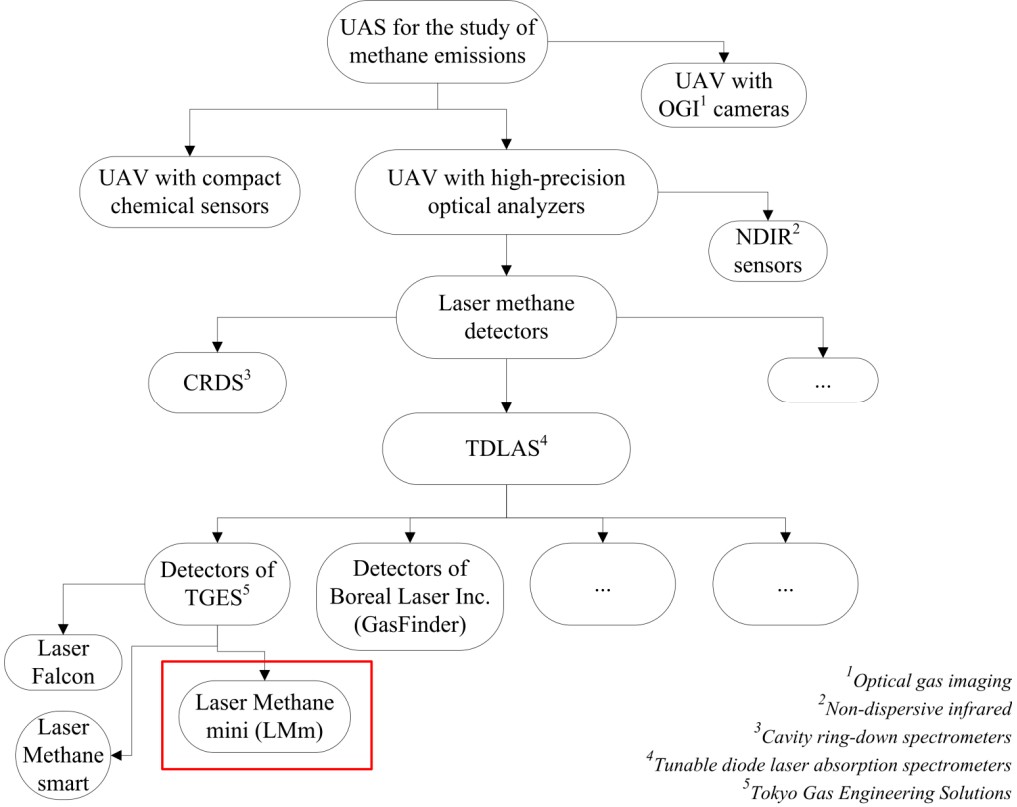

**Figure 1.** The location of the studied methane analyzer in a variety of other payload types.

This project used the LMm (SA3C32A) detector, more specifically the Bluetooth-enabled model LMm-BE (SA3C32A-BE), with an upgrade from the supplier (JSC Pergam-Engineering) to include a LEMO FGG.0B.305 output connector.

### 2.2. Selection of a UAV

In recent years, subject matter experts have noted that it is generally common practice to use multicopter UAVs for the study of methane emissions [12,13]. This can be explained by the fact that objects examined for methane leakages are usually fairly small (up to 1 km$^2$); in such conditions, multicopters, with their rotating wings, are far more advantageous than aircraft-type UAVs. They offer convenience and efficiency in setup and take-off, relative ease of landing, high maneuverability and the ability to hover over points of interest. When using multicopters, the spatial resolution of measurements can be increased thanks to the ability of this type of UAV to travel at slower speeds. For the end user, another factor to consider is that multicopters are, on average, cheaper than aircraft-type UAVs. Additionally, quadcopters are easier to fly consistently at low altitudes within one's line of sight, which can be an important consideration for remaining in conformity with flight regulations.

Aircraft-type UAVs are used in a minority of cases, mainly to study extended linear objects, such as natural gas transmission pipelines, where it is beneficial to make use of their higher performance and significantly greater flight range.

Among multicopters, quadcopters or hexacopters are the standard choices. The latter are naturally more expensive, but they have a higher carrying capacity and can therefore transport a wider range of payloads.

In accordance with the focus of this study on identifying a commercially available solution for monitoring methane emissions, it was decided to use an experimental UAV based on an open architecture and normally used by engineers—the X-FLY quadcopter manufactured by Unmanned Aviation Systems LLC (Russia, Perm). This UAV was used in combination with a Pixhawk Cube Black flight controller based on Arducopter 4.0.5 firmware and equipped with a U-blox NEO-M8P dual-system GNSS receiver with RTK capabilities. The UAV measured 65 cm in the diagonal and was designed with a fixed carbon-fiber landing frame and four arms equipped with brushless motors and 15-inch propellers. The power sources on board were lithium polymer 4S batteries with a capacity of 16 Ah (approximately 240 Wh). The assembly cost of the X-FLY UAV (excluding the payload) was approximately USD 650. The mass of the fully assembled UAV, including the battery but excluding the payload, was 3.0 kg. The flight time was 30 min with the selected payload (weighing 0.5 kg), at a temperature of 20 °C, and with a light wind of about 2–3 m/s. The LMm detector was mounted on the UAV using a 3D-printed plastic mount attached to the landing frame under the base. The X-FLY quadcopter with the LMm detector installed is shown in Figure 2.

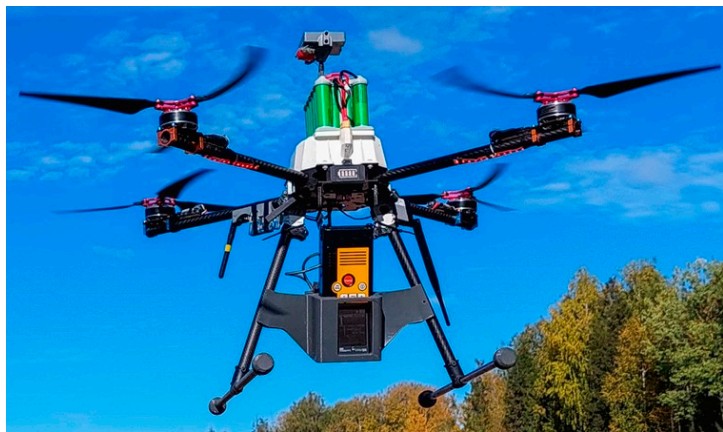

**Figure 2.** X-FLY Quadcopter with an LMm detector.

Flight tasks for gas surveys and UAV setup were both performed using the Mission Planner program (version 1.3.79), which is open-source software developed by Michael Oborne. If using commercial UAVs, such as those made by DJI, it would have been necessary to use a separate commercial setup program, such as DJI Assistant 2 for Matrice, as well as a separate flight assignment program designed for DJI UAVs.

### 2.3. Test Objects for the UAS

The proposed technical solutions were tested from May to September 2022 during a series of gas surveys completed at waste disposal sites (WDS) in the Perm region of Russia. Surveys were carried out on two landfills and two municipal solid waste (MSW) dumps; for the landfills, surveys were conducted once, and for the dump sites, they were conducted repeatedly. The main objectives of these surveys were to obtain maps of methane concentrations in the surface air layer above the objects, and to search for sources of methane emissions. The area of the objects surveyed at this stage varied from several hectares to several dozens of hectares.

The UgCS SkyHub onboard computer hardware (Model 2), from SPH Engineering in Latvia, was chosen by the authors as the starting technical solution for this setup. It had been tested a short time earlier during three surveys in the summer of 2021, which were also conducted from an X-FLY UAV. The survey objects then were also a landfill and an MSW dump, and the same tasks were set: establishing methane concentration maps and identifying sources of methane emissions.

## 3. Results

### 3.1. Review of Existing Non-Serial Solutions

Since the aim of this project was, in fact, to determine how best to integrate laser methane detectors into UAS, one of the main tasks was to use the equipment the authors had selected to develop optimal and accessible integration methods, such as commercially available ones, in order to ensure that the resulting proposed methods were user-friendly and easy to put into practice.

UAS setups with laser methane detectors began to emerge as an experimental practice at the beginning of the previous decade. In the study [14], Italian researchers used a custom-made TDLAS sensor from Physical Sciences Inc. (Andover, MA, USA), in combination with a custom-made quadcopter. The UAV's flight controller was used to capture detector readings, correlate them with planned measurement points (obtained from a differential GNSS receiver) and altitude coordinates (obtained from a pressure sensor), and to then record all the information on an internal drive. The flight controller was built based on an ARM processor, and its software was created using the .NET Micro Framework environment. Data on methane concentrations (with the coordinates of measurement points) were then transmitted to the ground control station via a radio channel. Based on this UAS, Terra Sana Consultants Pty Ltd. (Brisbane, Australia) developed a similar system that was successfully tested at Australian waste disposal sites [15]. Thus far, these developments have unfortunately not progressed beyond the level of prototypes, which remain difficult for other users to reproduce.

In another study by Swedish authors [16], a modernized DJI Matrice 210 was used as the UAV, and a modernized Aeris MIRA Pico (San Francisco, CA, USA) (a mid-IR laser) was used as the methane detector. Since the study involved the use of a number of climate sensors, the company Sparv Embedded AB designed a logger to record the readings of all sensors, including the one for methane. The same company performed a system integration of all sensors, along with separate GNSS and pressure sensors. The details of the system integration were not described in the publication.

Although the Swedish researchers did not plan to use data from the built-in GNSS receiver of the UAV in normal mode, they could not do without the flight data recorded by the autopilot since balance calculations necessitated data on the speed and yaw angles of the UAV throughout the flight. Therefore, when processing the measurement results, the authors had to solve the problem of how to synchronize two data streams: one from the logger and the other from the UAV flight controller. This synchronization was based on atmospheric pressure data, which was measured both by the built-in pressure sensor of the DJI autopilot and by a separate pressure sensor from Sparv Embedded AB. Combined data streams from the laser methane detector and the UAV flight controller were also used in a few other projects; in those cases, the authors also had to solve the problem of data synchronization [7,17].

In several instances [4,17,18], data on methane concentrations from the LMm detector were transmitted via Bluetooth to a mobile device with an Android operational system. The GasViewer mobile app from Pergam-Suisse AG (Kilchberg, Switzerland) could then be used for receiving, processing and presenting data. Since data on the UAV's coordinates at the time of measuring concentrations were taken from the autopilot, which received the information from its built-in GNSS sensor [17], processing the field data required that two arrays of information from different devices be synchronized. In the study [17], the data

arrays were combined in Microsoft Excel spreadsheets according to the timestamps of events that were present in both arrays. When it was not required to localize measurements in space with a high degree of accuracy, the coordinates of the measurement sites determined by the GNSS sensor of a mobile device could be used [4,18], resulting in one data stream (and hence an accuracy reduction).

Apparently, in another version of a UAS, coordinate marks were not assigned to the measured methane values during post-flight data processing but rather directly on board the UAV, using a separate microprocessor [3]. However, the details of this technical solution were not disclosed in the publication.

Even more complex system integration options were implemented by a group of predominantly German researchers led by P. Neumann [19,20] and by a group of American researchers [1,2].

In the German-led project, an onboard computing unit (UDOO X86, SECO S.p.A., Arezzo, Italy) with a separate microcontroller board (small ARM Cortex-M4 (ARM Limited, Cambridge, UK) development board) was placed on board the UAV. In that way, it was possible for all the data from the UAV payload to be collected, decoded and recorded, along with the information gathered via the methane detector and the altimeter. Data transmitted by the flight controller were also incorporated. It should be noted that the methane detector (an LMm) was also significantly upgraded. Its Bluetooth module was replaced with a custom-made electronic board, which enabled the detector to be connected to the microcontroller board and the computing unit via a universal asynchronous receiver/transmitter protocol (UART) communication interface.

In the American project, a commercial methane detector based on TDLAS technology was installed on a UAV (a custom-made quadcopter) after undergoing deep customization. In addition to replacing the laser, radically miniaturizing the electronic board and making other significant changes, a GNSS sensor was integrated into the methane detector for the purpose of georeferencing the measurement results [1].

A summary is provided in Table 1 to give an overview of the details involved in various studies focused on designing integrated systems between laser methane detectors and multi-rotor UAVs. For such projects, it has been common practice to select deeply modernized serial detectors or custom-made detectors. Furthermore, these system integration processes use complex hardware solutions, the details of which are usually not disclosed; consequently, the resulting solutions are difficult to reproduce or replicate, not to mention the fact that the complexity of these systems makes it necessary to place backup equipment such as microprocessors and computing units on board the UAV. However, simple hardware solutions, such as placing a smartphone on board to receive LMm readings via the Bluetooth channel, can lead to additional difficulties. These might be technical, such as the problem of securely fastening the smartphone on board the UAV, or they might be related to data post-processing and the need to synchronize the data stream from the methane detector with that of the UAV flight controller. Thus, the development of a simple and massively reproducible technical solution for the problem of system integration between a UAV and a methane detector continues to be a relevant challenge.

*3.2. An Example of a Serial Solution—The SkyHub System (SPH Engineering)*

There are several commercial serial solutions for building integrated systems around UAVs and laser methane detectors. One in particular was developed by the company SPH Engineering (Latvia, Riga) and is known as the SkyHub device, an onboard computer that is part of a software and hardware complex with the same name. It was designed to integrate various payloads, that is to say, various sensors (including laser methane detectors) and UAVs, into a single complex. A SkyHub connected to an LMm is shown in Figure 3.

**Table 1.** Example options for system integration of laser methane detectors and multi-rotor UAVs.

| UAV (Brand, Type) | Autopilot | Methane Detector | Detector Modification | Detector Mounting Method | Additional on-Board Equipment | Method for Combining Methane Detector Data with the Coordinates of Measurements | Research |
|---|---|---|---|---|---|---|---|
| Custom-made (quadcopter) | Custom-made (ARM processor based) | Experimental TDLAS-detector | N/A | No data | No | Via the flight controller | [14] |
| Custom-made (hexacopter) | Pixhawk | LMm | No | General circuit board | microprocessor, Android device | Methane detector data were transmitted via Bluetooth to an Android device and georeferenced using a separate onboard microprocessor | [3] |
| DJI Spread Wings S1000 (octocopter) | DJI WooKong—M | LMm | No | Aluminum mounting plate | Smartphone | Methane detector data were transmitted via Bluetooth to an Android smartphone and combined with flight controller GNSS data during post-processing | [17] |
| Custom-made (quadcopter) | No data | Remote Methane Laser Detector | Customization (including GNSS sensor integration) | No data | No | Via the methane detector | [1] |
| 3DR Solo (quadcopter) | Pixhawk 2 | LMm | No | Vibration-dampening 3D-printed plastic mount | Android device | Methane detector data were transmitted via Bluetooth to an Android device, where it was combined with GPS data built into the Android device | [4] |
| DJI Matrice 210 (quadcopter) | No data | Aeris MIRA Pico | Deep customization | No data | GNSS sensor, logger | Via the logger | [16] |
| DJI Matrice 600 Pro (hexacopter) | DJI A3 Pro | LMm | No | No data | Smartphone | Methane detector data transmitted via Bluetooth to an Android device, combined with inbuilt GPS data | [18] |
| DJI Spreading Wings S1000 (octocopter) | DJI A3 Pro | LMm | Deep customization | DJI Zenmuse Z15-A7 upgraded gimbal | Computing unit, microcontroller board, altimeter | Data were transmitted via UART interfaces, then combined using a computing unit and a microcontroller | [19,20] |
| DJI Matrice 600 Pro (hexacopter) | DJI A3 Pro | Experimental QCLAS * | N/A | Fixed frame mount | RTK-GNSS-sensor | RTK-GNSS sensor data was combined with detector data during post-processing | [7] |

* QCLAS—quantum cascade laser absorption spectrometer.

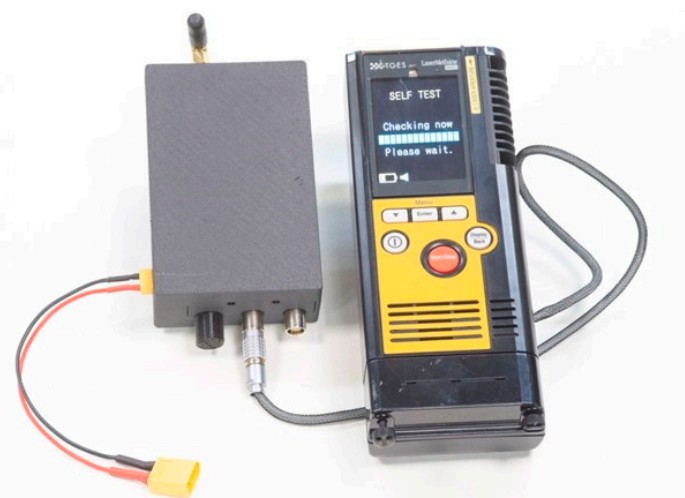

**Figure 3.** The SkyHub device with an LMm detector.

The SkyHub device is what served as a prototype for the technical solution implemented in this project. Documentation for the device is publicly available on the manufacturer's website [21]. A schematic diagram of an LMm and X-FLY UAV integration using SkyHub is shown in Figure 4.

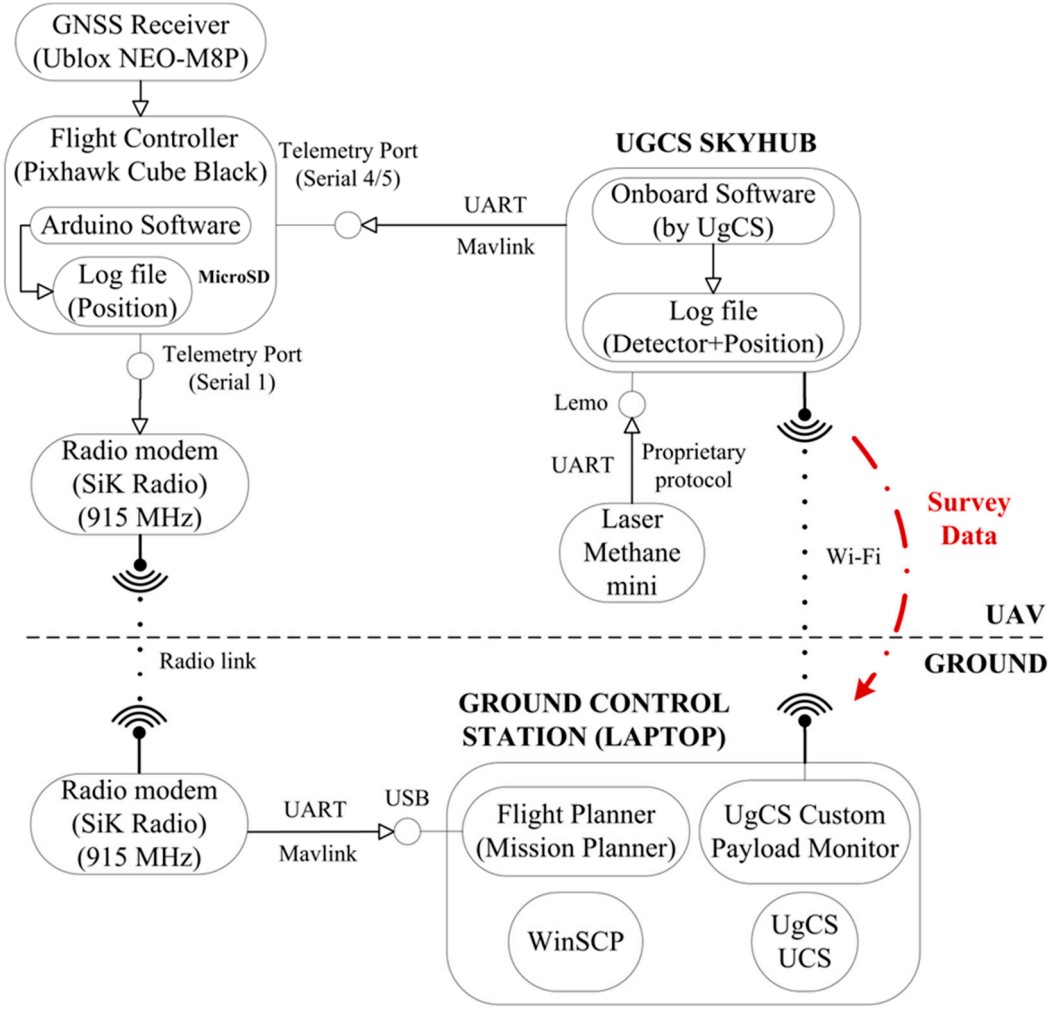

**Figure 4.** Integration scheme via SkyHub for an X-FLY UAV based on Pixhawk and ArduPilot with an LMm detector.

The main function of SkyHub in this case is to receive data both from an external payload (methane detector) and from the UAV flight controller, then to convert and record the data in a format convenient for processing and analysis [21].

The main component of the SkyHub device is a single-board microcomputer, which performs data reception and transmission, and signal conversion. The SkyHub device uses a proprietary protocol to receive data on exact methane concentrations from the LMm via the UART interface. From the UAV flight controller, as well as via the UART interface (using the Mavlink protocol), SkyHub receives the second data stream, which consists of data on the three coordinates of the UAV in space at the time of each methane concentration measurement. These two data streams are synchronized, combined and written to the SkyHub device's built-in memory. SkyHub also sends data on methane concentration measurements taken by the LMm to the flight controller via the UART interface. These data are then transmitted via a radio modem to a workstation in real time using the so-called "downlink" function.

The software aspect of the SkyHub complex includes a number of specialized programs and add-ons, some of them commercially available:

- UgCS Custom Payload Monitor (SPH Engineering)—for setting up aerial gas survey capabilities and obtaining data on methane concentration measurements in real time;
- WinSCP (developed by Martin Prikryl)—an open-source file management program for downloading data from SkyHub after the flight via Wi-Fi;
- LMC Process (JSC Pergam-Engineering)—for reading and primary processing gas survey data.

Analyzing the operational scheme of the complex, the authors noted that SkyHub plays a central role in the entire UAS as it is used to download flight data, conduct a two-way exchange of information with the UAV flight controller, and much more. This partly explains the high cost of a system integration option based on SkyHub. Given the price offers of both SPH Engineering itself, and their dealer in Russia, Moscow (JSC Pergam-Engineering), the SkyHub hardware and software complex increases the cost of LMm and Laser Falcon detectors by 150–200%. Therefore, a simpler and more logical solution would likely be to integrate the LMm directly into the flight controller and receive all flight data from the onboard computer, which would function as a sort of "brain" for the UAV.

Consideration of the SkyHub hardware and software complex as an example of an existing serial solution to the system integration problem confirmed our conclusion that it could be a highly worthwhile endeavor to develop an affordable and easy-to-execute scheme for integrating a laser methane detector with a UAV, using a method that would be readily accessible for the end user. Therefore, the main objective of this study was narrowed to focus on the development of a device similar to SkyHub but less expensive and easier to use, and without the functions of onboard flight data collection and processing, which would be left to the flight controller.

*3.3. Research Results*

To solve the problem of integrating a multi-rotor UAV and a laser methane detector, a software–hardware interface was created to transmit methane concentration measurement data from the detector to the UAV in digital form.

The hardware aspect of the device (the LaserHub+) is based on the Arduino Nano 5v processor. Connecting this processor to the hardware UART interface of the laser detector (19,200 baud rate) calls for a bidirectional logic level converter (5.5/3.3 V). The LaserHub+ also includes a power stabilizer (5/3.3 V) with a capacitor. A bidirectional logic level converter is also used to connect the Arduino Nano 5v to the Pixhawk Cube Black flight controller via a software UART interface (115,200 baud rate). The internal structure of the LaserHub+ is shown in Figure 5.

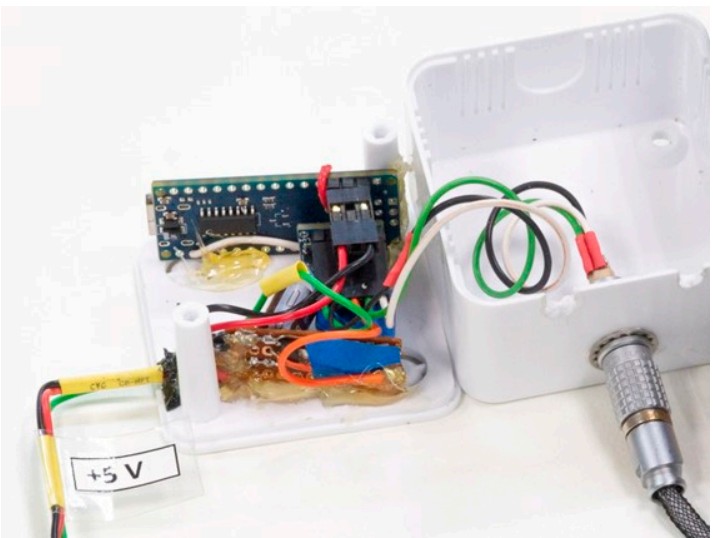

**Figure 5.** Internal structure of the LaserHub+.

The measurement data format received from the laser detector is converted by the LaserHub+ into a format accepted by the Pixhawk flight controller, with Arducopter code as the distance sensor or rangefinder data format. This data conversion (decryption of the data protocol closed by the LMm detector manufacturer) is made possible through reverse engineering methods. Thus, to send decoded data to the flight controller, the operation of a laser altimeter is emulated in accordance with the instructions [22], which were developed for the TF02 lidar (Benewake Co., Ltd., Beijing, China). The laser detector is connected to the flight controller via the Serial 4/5 Pixhawk port. The hardware design of the LMm and the Pixhawk flight controller integration via LaserHub+ is shown in Figure 6.

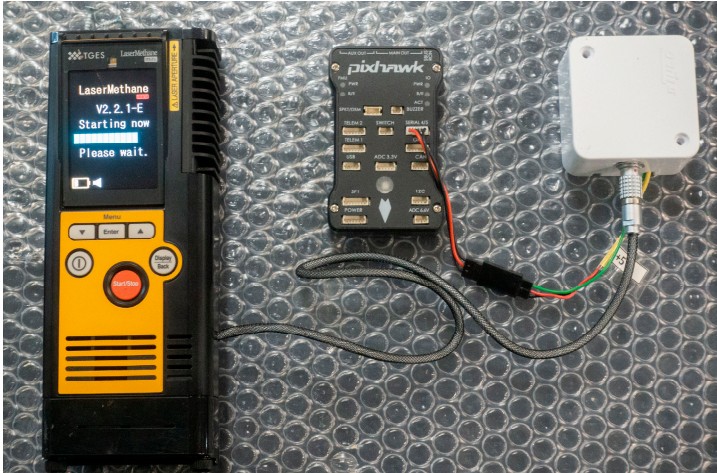

**Figure 6.** Connecting the LMm detector to the Pixhawk flight controller via the LaserHub+ device (on the left); LaserHub+ internals (on the right).

The flight controller receives methane concentration measurements, displays them in real time via the Mission Planner interface (Figure 7), and records them to a flight log file on removable media. Therefore, a downlink function, provided by Mission Planner, is used to display data on the operator's screen in real time which is similar to the approach taken by SPH Engineering in the SkyHub complex. A schematic diagram of an LMm and an X-FLY UAV integration using LaserHub+ is shown in Figure 8.

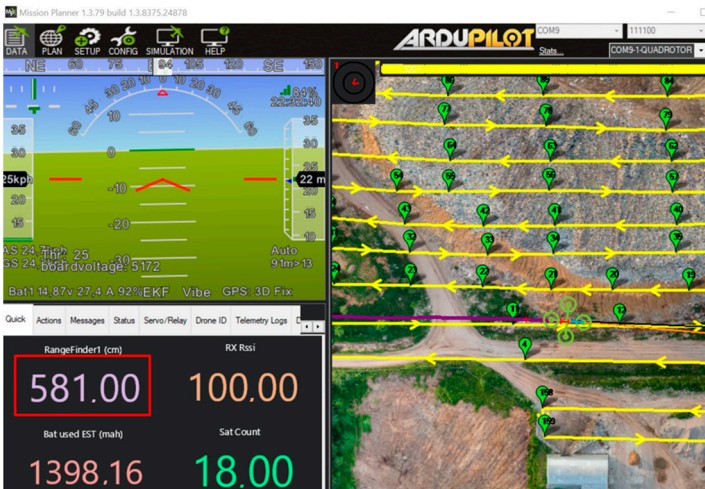

**Figure 7.** Display of the LMm detector readings in the Mission Planner program interface (highlighted with a red frame). The operation of the laser altimeter is emulated; the readings of the methane detector with a dimension of ppm*m are denoted with a dimension in cm.

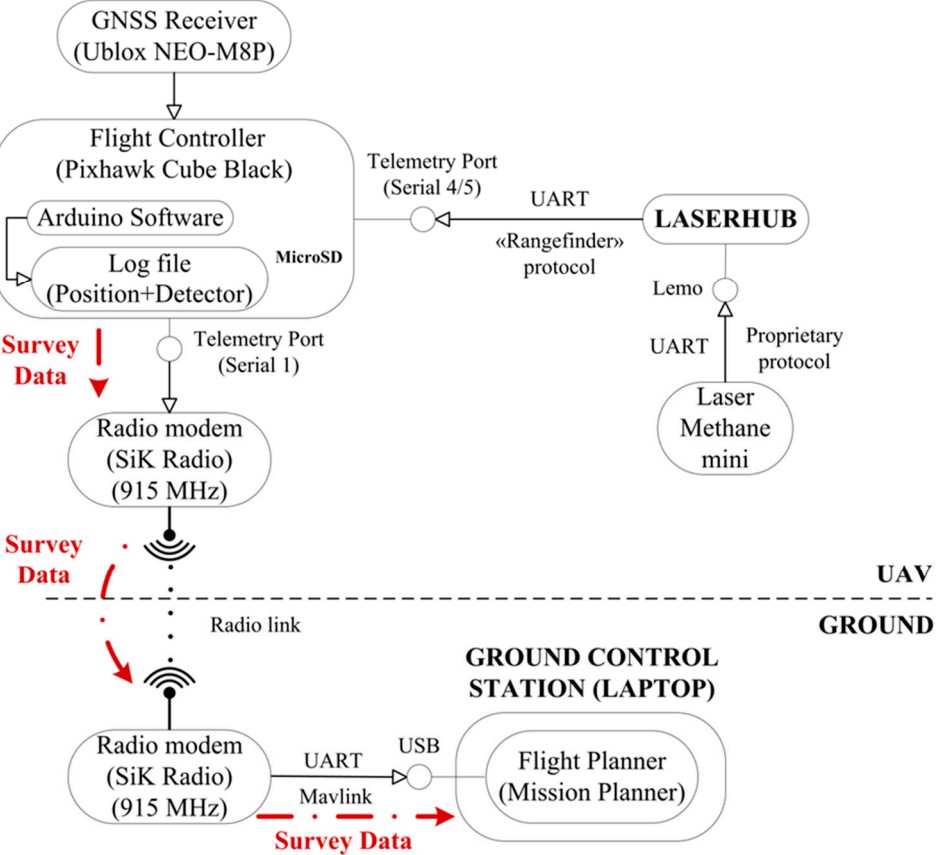

**Figure 8.** Integration scheme via LaserHub+ of an X-FLY UAV based on Pixhawk and ArduPilot and the LMm detector.

The measurement data recorded in the log file are labeled as RFND (rangefinder). To process the log file data, it is recommended to convert the file from the .bin format to the .log format and to work with the RFND data in a spreadsheet editor, such as Microsoft Excel, so as to more easily compare it with the coordinates of the measurement points, which are labeled as "GPS". When comparing these data groups against the flight log entry numbers, one should take into account the different frequency at which measurements are performed by the onboard GNSS receiver versus by the methane detector. The

operating frequency of the LMm measurements is 10 Hz, whereas the operating frequency of the onboard GNSS receiver is 5 Hz. Consequently, there are several (usually two) measurements of methane concentrations per one measurement of coordinates, so they must be averaged, which can be easily performed with a table editor. In detail, the process is as follows: each coordinate measurement that has its own flight log entry number is compared with the average of two methane concentration measurements that have their own entry numbers that are closest to the flight log entry number of the coordinate measurement taken. Accordingly, the accuracy of the synchronization of the methane concentration measurements and the coordinates of the UAV position at the time of the measurements is within one measurement period of the methane detector, i.e., within 0.1 s (at its operating frequency of 10 Hz).

However, the question of the delay between the moment when the gas concentration is measured by the methane detector and the transmission of these readings to the LaserHub+ remains insufficiently clear. To estimate the magnitude of this delay, laboratory tests on a test bench are required. We assume that this delay is of the same order (0.1 s).

To set up the methane leakage detection complex, the end user needs to utilize the installed connectors to connect the LaserHub+ to the flight controller and to the laser detector placed on board the UAV. It is also necessary to set up the UAV autopilot in the Mission Planner software by entering the following parameters under the tab "Full Parameter List":

- SERIAL4_PROTOCOL = 9 (Lidar);
- SERIAL4_BAUD = 115 (115,200 baud);
- RNGFND1_TYPE = 20 (Benewake-Serial);
- RNGFND1_MIN_CM = 0.

The following protocol is implemented for an interaction with the gas analyzer:

1. Submit a data request to the LMm. A 13-byte string is used as a request (it is sent at a frequency of 2 Hz):

$$\text{x «\{\textbackslash x02\}ETC:FWD ?; \{\textbackslash x03\}\{\textbackslash x26\}»}$$

2. Wait for a response from the LMm. The response line (295 bytes) must begin with the sequence:

$$\text{«\{\textbackslash x02\}ETC:FWD»}$$

3. In the response line, read bytes 47–51 containing the desired value.
4. Convert the read data to a numeric type.

To send the read data to the flight controller, they are prepared by means of a series of calculations:

1. The top byte for the send array is obtained:

$$\text{TopByte = (GasAnalyzerValue >> 8) \& 0b11111111}$$

2. The low byte for the send array is calculated:

$$\text{LowByte = GasAnalyzerValue \& 0b11111111}$$

3. The check byte is calculated:

$$\text{CheckByte = (0x59 + 0x59 + TopByte + LowByte) \& 0b11111111}$$

4. An array (9 bytes) is formed to be sent to the autopilot:

Array [0]: 0x59
Array [1]: 0x59
Array [2]: LowByte
Array [3]: TopByte
Array [4]: 0x00
Array [5]: 0x00
Array [6]: 0x00
Array [7]: 0x00
Array [8]: CheckByte

5.  The formed array is sent to the autopilot at a frequency of 100 Hz.

The .ico file format with the source code to be downloaded to the Arduino Nano is given in the Supplementary Materials.

To assemble the entire airborne methane leakage detection system, one must acquire a mount to attach the LMm to the body of the UAV; the LaserHub+ casing, which, unlike the rest of the components, cannot be bought as a ready-made product, is also a necessity. DJI devices are generally sent from the supplier with a kit that includes special plastic brackets for installing SkyHub. In our project, however, the mount for the LMm and LaserHub+ and its case were printed on a 3D printer according to original drawings made using AutoDesk Inventor software, version 2021 (drawings in .stl format files are given in the Supplementary Materials).

The mount for the LMm and LaserHub+ can be seen in Figure 9. There are a variety of options for mounting the payload on a UAV body, some of which are very simple. The main criteria to bear in mind are the correct orientation of the laser methane detector and the reliability of the fasteners.

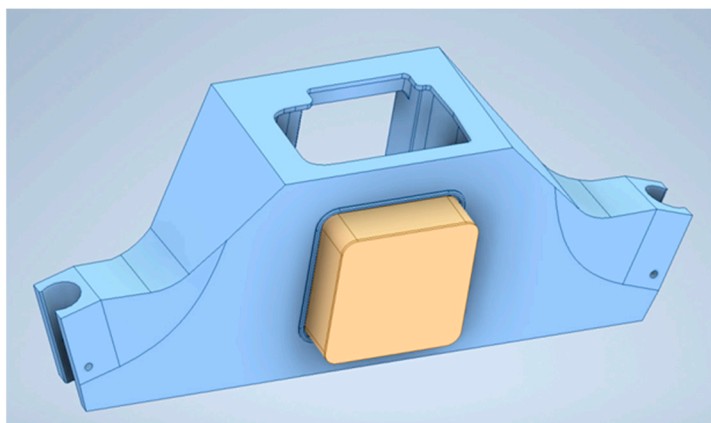

**Figure 9.** Plastic mount for LMm detector (blue) and LaserHub+ (yellow) for installation on an X-FLY UAV.

The LaserHub+ device was successfully tested in the field during gas surveys of MSW sites in the Perm region of Russia (see SubSection 2.3). It was confirmed during these surveys that the results of measuring methane concentrations, along with the corresponding coordinates of the measurement points, are indeed transmitted from the UAV to the ground control station in real time, and recorded in the flight controller log file on a removable storage device (in this case a memory card). Gas survey data from the log file are extracted using Microsoft Excel and can then be analyzed in specialized programs, a geographic information system (GIS), for example.

When using the UAV with LaserHub+ on board, we were able to avoid several difficulties we had faced with SkyHub. These were, most notably, due to the fact that the data stream received from the SkyHub device to the workstation, using WinSCP software (version 5.17.7), did not contain data on the altitude coordinates of the points where the methane concentration measurements had been taken. According to the technical support

services of the manufacturer (SPH Engineering) in June–July 2021, transferring altitude coordinates from Pixhawk flight controllers to SkyHub was an ongoing problem. To resolve this issue, we therefore had to use data from the log file of the Pixhawk flight controller and subsequently synchronize them with data from the log files received from SkyHub, thereby making it possible to restore the altitude marks of the measurement points necessary for the correct processing of the gas survey results (in particular, for converting measured methane concentrations from ppm-m to ppm). At the time of publishing this study, exact information as to whether this issue has been solved is not available.

## 4. Discussion

The developed LaserHub+ module, which solves the problem of integrating a laser methane detector and a UAV into a single monitoring complex for detecting methane leakages, has a number of undeniable advantages over its predecessor: the SkyHub device by SPH Engineering. And yet, it should be said that SkyHub has a much wider application than LaserHub+, which therefore enables it to integrate a greater variety of payloads (laser methane detectors, altimeters, anemometers, metal detectors, etc.) into research complexes with UAVs based on various flight controllers (DJI, ArduPilot and PX4). However, for solving a particular problem, such as the search for and study of methane emissions sources, LaserHub+ was shown to be a far more preferable solution (more convenient, more practicable and more economical) than SkyHub (Table 2).

In particular, the noticeably smaller mass and lower power consumption of the LaserHub+ device leads to an increase in the potential flight time of the UAV and, accordingly, an increase in the performance of the UAS. The absence of the need to use a number of specialized software programs not only simplifies the configuration and use of the entire unmanned complex but also greatly reduces the development cost. For example, according to Table 2, the fees for software and licenses when using SkyHub represent more than 60% of the entire software and hardware complex cost. The LaserHub+ module also has advantages over existing experimental setups (Table 1), since it is a simpler product that end users can easily assemble independently from affordable and readily available components. LaserHub+ users only need one free public software (Mission Planner) to set up equipment, plan flights (for gas surveys) and extract the collected data.

**Table 2.** Comparison of SkyHub and LaserHub+ devices.

| Parameter | SkyHub | LaserHub+ |
|---|---|---|
| Weight [1], g | 200 | 49 [2] |
| Dimensions, L × W × H, mm | 109 × 69 × 34 | 54 × 54 × 30 |
| Power consumption [3], W | 1.7 (3.0 [4]) | 0.1 |
| Required software | UgCS SkyHub (onboard software), UgCS UCS, UgCS Custom Payload Monitor, Mission Planner | Mission Planner |
| Cost for hardware, € | 2560 [5] | 25 |
| Cost for software (incl. licenses), € | 4340 [5] | Not applicable [6] |

[1] Without fasteners; [2] Together with an output cable for connecting to the UART interface of the flight controller; [3] Under laboratory conditions (no payload); [4] Maximum value according to the manufacturer's documentation; [5] Cost is indicated excluding taxes and fees according to the SPH Engineering website [23]; [6] Free of charge to the end user, as the equipment is configured in the free Mission Planner environment.

The software configuration of the LaserHub+ setup is quite simple, so there is no need for any specialized software that may or may not be commercially available. In comparison, the SkyHub set up requires elaborate configuration operations, and other complexes

require original software (see examples in Table 1). Connecting a laser methane detector to a flight controller via LaserHub+ is very simple and can be performed quickly; the only potential difficulty that may arise is equipping the LMm with a LEMO connector.

There is also some difficulty in designing and manufacturing a mount for a methane detector; however, this complexity is common to all custom-made UAVs based on Pixhawk and ArduPilot because there are no compatible serial mounts on the market, a fact that is not expected to change anytime soon. Assuming the presence of a methane detector securely attached to the UAV, the LaserHub+ device itself is quite easy to install with the help of elementary improvised means, thanks to its light weight and small dimensions.

Aside from SkyHub, there are a few other serial commercial solutions that make it possible to combine data on measured methane concentrations with data on measurement point coordinates. For example, JSC Pergam-Engineering (Russia) supplies customers with data loggers of its own design, together with laser detectors manufactured by GASTAR Co. Ltd. (Japan), which can be used to georeference the results of methane concentration measurements. These devices were not studied as part of this project, but it should be noted that their cost is much higher than that of LaserHub+ and they lack the capacity to transmit data to the UAV ground control station in real time.

An important factor to consider is how widely the proposed device can be applied. The technical solutions implemented in this study are applicable to other multicopter UAVs based on Pixhawk flight controllers. That said, it is possible that other laser detectors from GASTAR Co. Ltd., especially the Laser Falcon, could also be combined with UAVs in a similar manner. This assumption is based on the fact that the LMm and Laser Falcon detectors both have: (a) similar schemes for configuration of and connection to SkyHub, and (b) similar operating principles. Analysis of the scientific literature shows that the LMm and Laser Falcon are among the most common choices when conducting aerial gas surveys to study methane emissions. It follows, therefore, that our LaserHub+ device could one day be in great demand, especially since it relies on open rather than commercial hardware and software.

## 5. Conclusions

Studying emissions of methane, which is one of the most common atmospheric pollutants and a powerful greenhouse gas, requires the use of modern methods and technical means for measuring the concentrations of the target gas in the environment. One such modern approach is airborne laser absorption spectroscopy, performed using portable methane detectors mounted on board UAVs. At the same time, it is essential to remind ourselves that these resources must remain accessible, convenient and straightforward for the end user.

The proposed development—the LaserHub+ device—makes it relatively easy to integrate one of the most widely available methane detectors, the Laser Methane mini, into an unmanned aerial remote-sensing complex, based on a multi-rotor UAV built on the Arduino platform with a Pixhawk flight controller. The LaserHub+ then serves as a decoding device that transmits data from the LMm to the flight controller. Such an integration scheme takes a more logical approach of assigning the central role in data collection and processing to the UAV flight controller, so there is no need to deeply modernize serial methane detectors, use custom-made detectors with complex control electronics, place additional devices (smartphones, GNSS sensors, etc.) on board the UAV, or apply any other hard-to-replicate solutions.

Over the course of this project, we examined the pre-existing experimental solutions described in other studies and implemented them as custom-made UAVs, taking the SkyHub by SPH Engineering as our main representative sample for comparison. The LaserHub+ device clearly represents a solution that is less costly, much simpler and more convenient to use than similar serial commercial solutions available nowadays. This publication contains all the data needed by an end user to implement the proposed technical

solutions, which constitute an important contribution to the development of research into methane emissions in various industries and sectors of the economy.

**Supplementary Materials:** The following supporting information can be downloaded at: https://www.mdpi.com/article/10.3390/drones7100625/s1, The .ico file format with the source code to be downloaded to the Arduino Nano; Drawings in .stl format files.

**Author Contributions:** Conceptualization, N.S. and T.F.; methodology, T.F. and I.L.; validation, T.F. and I.L.; investigation, T.F. and I.L.; resources, N.S.; data curation, T.F. and I.L.; writing—original draft preparation, T.F. and I.L.; writing—review and editing, N.S.; visualization, T.F. and I.L.; supervision, N.S.; project administration, N.S.; funding acquisition, N.S. All authors have read and agreed to the published version of the manuscript.

**Funding:** This study was performed with financial support from the Ministry of Science and Higher Education of the Russian Federation (Project No. FSNM-2020-0024).

**Data Availability Statement:** Not applicable.

**Acknowledgments:** The authors would like to thank the engineer A.V. Zimin, who played a leading role in the development of the LaserHub+ hardware interface and the LaserHub+ interaction protocols with the LMm detector and the Pixhawk flight controller.

**Conflicts of Interest:** The authors declare no conflicts of interest.

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
