# Peer review of "Integrating a UAV System Based on Pixhawk with a Laser Methane Mini Detector to Study Methane Emissions"

_drones, doi:10.3390/drones7100625_

Round 1

Reviewer 1 Report

Idea is good and significantly simplifies transmission of gas concentration data in the specific described circumstances.

Focus on reproducibility, low-cost, and use of open-source software is good.

Good background/intro/review/overview part.

Leads to a somewhat short result part which could be more detailed (part and supplier identification of used components, source code of developed software, actual achieved performance).

Result part has a bit of an unfinished impression, might benefit from some restructuring/proofreading.

The mentioned figure of the mount is missing (459).

Especially since the authors criticize in other papers a lack of details for reproducibility, more details on the implementation and a public repository of the source code (e.g. github), or an appendix of the source code would be beneficial.

Same is true for the wiring/electrical diagram, even though this is somewhat trivial.

But in general, enough details to reproduce with some minor efforts are given.

The authors make no remark on the actual delay/error achieved in synchronization due to delay in the sensor itself or serial data transmission.

An actual estimate or measurement of the shift between the measurement time and location time would be interesting and if not possible should at least be mentioned.

Some parts could benefit from additional proofreading/spellchecking, see remarks below.

Some minor remarks are given on a per-line basis below.

33: 'a' missing

96: 'to' missing

117: verb missing

129-136: Missing second wavelength/wavelength tuning aspect for reference measurement.

142: 'of' missing?

165: unclear what "and a payload equipped with universal asynchronous receiver/transmitter protocol" refers to

174: what is meant by "spatial frequency measurements can be increased" is unclear

182: 'productivity': wrong word?

191: double space

194: 16 mAh dubious. 16 Ah?

197: flight *time*?

204: quotation marks missing for software name

274: the -> an?

303: that->the

321. Spread Wings -> Spreading Wings

326: Figure 3 caption missing

387: Sentence could be restructured to be more clear

389: decrypted is wrong -> decoded or parsed data

400: display data function: no details on the implemented display function, explanation or screenshot would be nice

400: might be misleading: was the display function implemented or simply the Mission Planner functions used? If the last: More clearly state that this part is already implemented and provided by mission planner/ardupilot.

435: verb missing

459: Image/Figure of the mount missing

534: space missing

549/550: naming in brackets confusing, maybe move to single bracket, unclear if this is the intended position for that information, unclear what the difference between software and programs is

Further minor remarks:

LaserHub vs. LaserHub+ is used inconsistently.

UAS vs. UAV is used inconsistently.

Use of the words 'engineering sample' is sometimes confusing. Maybe 'prototypes' is what is meant?

Sometimes unclear what is meant by 'serial' in the context of products.

Some parts could benefit from additional proofreading/spellchecking, see remarks.

Author Response

The authors would like to thank all the reviewers for their valuable comments. Major changes have been made

Please find the response to reviewer comments in the attached file

Reviewer 2 Report

This article successfully integrates methane sensing and drones, which can contribute to related environmental monitoring applications. The following suggestions are provided for reference:

1. Many hardware integration processes and details have been explained in the article, but there is no explanation for the presentation of the collected data, especially whether the monitored data is compared with other monitoring data in the field to prove the subsequent practical application. In addition, if the data is successfully displayed in GIS and compared with the verification data, the value of the article will be greatly improved.

2. line 323, the title of sec. 3.2 should be move to the back of Figure 3.

3. P.8, Table 1, It is recommended to adjust the text size and description of the table content and try to present the complete content on one page. It is also recommended that the results of this study be directly included in the table for comparison, so as to highlight the contribution and improvement of this study.

4. P.10 and P.14, Figure 4 and Figure 7, It seems that these two pictures can be integrated to make the article more readable.

5. line 459, "The mount for the LMm can be seen in Figure 7. " The article description does not match the picture title and content.

6. SkyHub / UgCS all SPH Engineering's products, In addition to the integration of flight control and methane sensor hardware, this article explains whether there are other improvements? Otherwise, there will be relevant research results in 2019...

Author Response

(The authors gave the same response as above.)
